# Aptamers as Theragnostic Tools in Prostate Cancer

**DOI:** 10.3390/biom12081056

**Published:** 2022-07-29

**Authors:** Carlos David Cruz-Hernández, Griselda Rodríguez-Martínez, Sergio A. Cortés-Ramírez, Miguel Morales-Pacheco, Marian Cruz-Burgos, Alberto Losada-García, Juan Pablo Reyes-Grajeda, Imelda González-Ramírez, Vanessa González-Covarrubias, Ignacio Camacho-Arroyo, Marco Cerbón, Mauricio Rodríguez-Dorantes

**Affiliations:** 1Laboratorio de Oncogenómica, Instituto Nacional de Medicina Genómica (INMEGEN), Mexico City 14610, Mexico; carlosdavidcruzher@gmail.com (C.D.C.-H.); griseldargzmtz123@gmail.com (G.R.-M.); sergio.cortesram@gmail.com (S.A.C.-R.); mp.miguelmorales@gmail.com (M.M.-P.); marian.cruz.bqd14@outlook.com (M.C.-B.); correo.garciabeto@gmail.com (A.L.-G.); 2Laboratorio de Estructura de Proteínas, Instituto Nacional de Medicina Genómica (INMEGEN), Mexico City 14610, Mexico; jreyes@inmegen.gob.mx; 3Departamento de Atención a la Salud, Universidad Autónoma Metropolitana–Xochimilco, Mexico City 04960, Mexico; imeldagr14@gmail.com; 4Laboratorio de Farmacogenómica, Instituto Nacional de Medicina Genómica (INMEGEN), Mexico City 14610, Mexico; vgonzalez@inmegen.gob.mx; 5Unidad de Investigación en Reproducción Humana, Instituto Nacional de Perinatología, Facultad de Química, Universidad Nacional Autónoma de México (UNAM), Mexico City 04510, Mexico; camachoarroyo@gmail.com (I.C.-A.); mcerbon85@yahoo.com.mx (M.C.)

**Keywords:** prostate cancer, aptamers, Prostate-Specific Membrane Antigen (PSMA), Prostate-Specific Antigen (PSA), PCa diagnosis, PCa treatment

## Abstract

Despite of the capacity that several drugs have for specific inhibition of the androgen receptor (AR), in most cases, PCa progresses to an androgen-independent stage. In this context, the development of new targeted therapies for prostate cancer (PCa) has remained as a challenge. To overcome this issue, new tools, based on nucleic acids technology, have been developed. Aptamers are small oligonucleotides with a three-dimensional structure capable of interacting with practically any desired target, even large targets such as mammalian cells or viruses. Recently, aptamers have been studied for treatment and detection of many diseases including cancer. In PCa, numerous works have reported their use in the development of new approaches in diagnostics and treatment strategies. Aptamers have been joined with drugs or other specific molecules such as silencing RNAs (aptamer–siRNA chimeras) to specifically reduce the expression of oncogenes in PCa cells. Even though these studies have shown good results in the early stages, more research is still needed to demonstrate the clinical value of aptamers in PCa. The aim of this review was to compile the existing scientific literature regarding the use of aptamers in PCa in both diagnosis and treatment studies. Since Prostate-Specific Membrane Antigen (PSMA) aptamers are the most studied type of aptamers in this field, special emphasis was given to these aptamers.

## 1. Introduction

For decades, the efforts to understand and cure diseases have been a challenge in medicine. Although many drugs have been developed to treat specific diseases, the effect of these drugs is still dependent on many factors including genetic, environmental, geographical, clinical, etc. In cancer, several drugs have been used to treat each type of cancer; however, cancer cells develop mechanisms to escape from the effect of such drugs, making a constant update of the current drugs and targets necessary. The diagnosis and treatment of cancer heavily relies on specific biomarkers that allow early detection and targeted therapeutic agents (drugs that specifically target cancer cells by the precise delivery of chemotherapeutics and decrease toxic effects to non-cancer cells). Antibodies and nucleic acid probes remain as the main molecules used to investigate, characterize, and develop new biomarkers [1]. A wide variety of tools ranging from small drugs to viral vectors have been used to provide highly specific cancer therapies based on molecular interactions [2]. However, despite their efficacy in vitro and in animal models, their therapeutic outcome is compromised by either their lack of effect in vivo or toxic side effects. Aptamers have emerged as a relatively new molecular tool to overcome these issues. Aptamers (from the Greek *Aptus*, adequate and *Merus*, particles) are RNA or DNA single-stranded oligonucleotides. These small molecules (ranging from 20 to 100 nucleotides roughly) can be designed for recognizing almost any target molecule [3]. Most of the aptamers are designed and selected from in vitro, in silico, or even in vivo selection using a process called SELEX (Systematic Evolution of the Ligand by Exponential Enrichment) [4]. These aptamers have been studied in prostate cancer (PCa) research, particularly the PSMA (Prostate-Specific Membrane Antigen)-derived aptamers. In this review, we addressed the main techniques used in the selection and design of aptamers and their current use in prostate cancer research, with a focus on aptamers targeting the PSMA protein used for diagnostic and therapeutic PCa research.

## 2. Aptamers

Aptamers are DNA and RNA oligonucleotides that can adopt tridimensional structures that enable them to join specifically to any desired target (Figure 1) [5]. Aptamers are capable of binding to specific molecules including drugs [6], proteins [7], carbohydrates [8], cells [9], and viruses [10]. Aptamers were first described in 1990, and since then several groups have used their binding properties to isolate a diversity of specific aptamers [11]. This interaction is typically strong with dissociation constants ranging at picomolar concentrations [12]. Aptamer shapes fold into a tertiary structure by pairing with their complementary bases and folding into itself through magnesium ion bonds between the non-helical regions and the phosphate backbone of the ribose at the 2′-OH site [13].

Aptamers are often compared to antibodies; however, they possess many additional advantages in biological and production aspects [14]. For example, compared to antibodies and given their size and nature, they are non-immunogenic and have a greater ability to penetrate tissues. In addition, aptamers do not require a living organism for their production; they can be produced in vitro and easily modified with functional groups or probes for conferring them more stability. Hence, their production in large quantities is simple and affordable due to both the newest and easiest tools for chemical synthesis and to the new mutant enzymes that allow producing modified aptamers using polymerase chain reaction (PCR). Aptamers are generally selected in vitro from a large single-stranded oligonucleotide library comprising more than 1 × 10^16^ molecules by several rounds of selection (generally 6–12 cycles), and then they can be modified for optimization depending on the desired applications [15]. Table 1 summarizes the main advantages of aptamers over antibodies. Therefore, the therapeutic applications of aptamers may be broad and could become a viable alternative to the traditional antibody therapies as they enable drug delivery, diagnosis, imaging, and even the discovery of new biomarkers.

Despite the many advantages of aptamers over antibodies, there are some disadvantages of aptamers to take into account that could represent a barrier for its clinical application. First, their small size makes them more prone to degradation by nucleases in serum, shortening their half-lives in the circulation [16,17]. Thus, downstream modifications are usually required before their use in vivo. Moreover, the successful rate of effective aptamer identification by conventional SELEX processes is usually low and they may fail to inhibit their targets in vivo [18]. Finally, aptamer modifications that could help to overcome these barriers have to be done carefully to avoid these modifications could affect the folding structures and lead to loss of function.

**Table 1 biomolecules-12-01056-t001:** Advantages of aptamers over antibodies.

Features	Description	Advantage	Reference
**Size**	6–30 kDa 20–100 nt	Aptamers can be better distributed in tissues Less immunogenic than antibodies	[19]
**Aptamer–antigen interaction**	Highly specific	Aptamers can distinguish between molecules with only one methyl group of difference	[20]
**Chemical modification**	Easy chemical transformation	Aptamers can be modified and conjugated to a variety of molecules such as fluorophores, nanoparticles, drugs, or siRNAs	[21]
**Thermal stability**	High resistance to denaturalization	Aptamers can be refolded to their specific 3D conformations after incubation at 65–95 °C	[21]
**Cost of production**	Low cost of production	Aptamers can be produced at a high scale by chemical synthesis	[22]

Despite the extensive research undertaken in the field of aptamers, to date, Pegaptanib is the only successful aptamer used in clinical applications. This cased-aptamer drug was approved by the FDA for its use in humans for the treatment of age-related macular degeneration (AMD) [23]. Pegaptanib (Macugen^®^) is an RNA aptamer that specifically binds to and blocks the activity of the 165 amino acid isoform of the vascular endothelial growth factor (VEGF_165_), the major inducer of abnormal blood vessel growth and leakage in wet AMD [24]. Moreover, there are several aptamers under different stages of clinical trials for the treatment of diseases such as COVID-19, macular degeneration, cancer, coagulation, and inflammation (Table 2). For example, ARC1779 (Archemix) for thrombotic micro-disease, AS1444-AGRO001 (Antisoma-Archemix) for acute myeloid leukemia, NOX-A12 (NOXXON Pharma) for the treatment of lymphoma, NOX-E36 (NOXXON Pharma) for type 2 diabetes, and diabetic nephropathy are aptamers that are in phase I and phase II clinical trials [25]. Table 2 describes the aptamers that are being studied in clinical trials for either diagnosis or therapy of several diseases. Additionally, numerous works are ongoing at preclinical trials showing its potential applicability in theragnostics.

## 3. Aptamer Design

The procedure for designing aptamers is based on the Systematic Evolution of Ligands by Exponential Enrichment (SELEX). SELEX is a powerful tool for aptamer design due to the final specificity of the aptamer to its target after artificially increasing the pressure selection.

### 3.1. SELEX

SELEX involves the progressive selection of APTAMER–TARGET complexes consisting of repeated rounds of partition and amplification. The process starts with a large DNA or RNA single-stranded oligonucleotide library harboring a variable sequence and known 5′ and 3′ sequences [4]. The library may contain ~10^15^ oligonucleotides possessing approximately ~20–60-nucleotides (nt) in length with enough structural diversity to identify many targets [26]. The SELEX process is composed of three steps: The first step involves a repeated incubation of the single-stranded oligonucleotide library with the target. In the second step, the selection of high-affinity from low-affinity aptamers occurs through diverse isolation methods, depending on the binding molecule, for example, competitive elution for sequences not associated with the target [27,28]. In the final step, aptamers with high affinity are amplified using traditional PCR to produce several copies. In the case of RNA aptamers, an additional transcription reaction will be necessary. The whole SELEX process is repeated until the high affinity aptamers are enriched, which in turn is evaluated by flow cytometry or new generation sequencing [29].

### 3.2. Variants of SELEX

Although SELEX is successful in the selection of high affinity aptamers, this advantage can turn into a disadvantage because at the end, the aptamer–target binding is only achieved under the high specific conditions used in the aptamer selection [29]. Thus, it is not surprising that aptamers developed against purified proteins are unable to recognize their targets in their native environment. Several modifications have been made to the SELEX method to improve the selection process for obtaining functional aptamers.

#### 3.2.1. In Silico SELEX

Computational methods in aptamer selection, i.e., in silico SELEX, have emerged as a novel resource to improve and increase the efficiency of the process. This method is based on computational chemistry and theoretical physicochemistry calculations to select sequences of nucleotides with the potential of binding to ligands with high affinity from a pool of random sequences, lowering investment and wet lab time [30]. Thus, in silico SELEX strategies have been used to design a starting pool of sequences of either DNA or RNA aptamers. In the case of RNA, in silico SELEX strategies have the advantages of reducing the RNA sequence search space in four to five orders of magnitude, decreasing the time spent in the experimental selection rounds and increasing the selection of high-affinity aptamers [31]. Table 3 (modified from [32]) summarizes the main in silico tools used in the SELEX process. New and powerful bioinformatic tools have been developed and used for improving RNA aptamer selection such as Discovery Studio 3.5 tools (ZDOCK and ZRANK) [33] and RaptRanker [34], which have become valuable tools that increase the efficiency of high throughput SELEX (HT-SELEX).

In the case of DNA aptamers, the SELEX process is followed by in silico docking strategies to find motifs, optimize mutations, and increase target affinity [35]. Hence, some tools have been developed to predict DNA–protein complexes and DNA ring formations to optimize DNA aptamer design and selection [36]. To summarize, in silico tools are helpful to optimize, rank, and select aptamers increasing their affinity, decreasing wet lab time and cost, and leading to a better performance of the SELEX strategy.

#### 3.2.2. Negative SELEX

When incubating aptamers with targets as proteins, drugs, small molecules, peptides, or nucleic acids, the molecules are often immobilized to a specific matrix. Thus, during the selection process, some of the sequences may unspecifically bind to the other reagents in the reaction mix, which may result in many false-positive results from nontarget aptamers. To decrease the false-positive yield, Ellington and Szostak designed negative SELEX, which consists of incubating the enriched aptamers with an agarose matrix and resins used for purification [37]. This method increases the affinity of aptamers up to 10 times, and most of the current SELEX strategies include a negative SELEX step to eliminate non-specific sequences [38].

#### 3.2.3. Counter SELEX

Counter SELEX is similar to negative SELEX, but instead of using agarose supports, this method exposes the library to similar target molecules. Counter SELEX was designed by Jenison, R. et al. (1994) to increase RNA aptamer’s affinity and specificity to the bronchodilator theophylline by incubating the aptamer library with caffeine and theophylline (both molecules differ only by a methyl group at nitrogen, N-7) [39]. This incubation produced a 10,000-fold increase in aptamer’s affinity for caffeine. Since counter SELEX uses analog targets for the selection process, the target affinity and the structural selectivity achieved is higher than that obtained by negative SELEX and counter SELEX.

#### 3.2.4. Photo-SELEX

In photo-SELEX, a modified nucleotide activated by light absorption is incorporated in place of a native RNA- or in ssDNA-randomized libraries [40]. This method was first reported by Jensen et al. (1995) to identify RNA sequences that bind with high affinity and crosslink to the Rev protein from the human immunodeficiency virus type 1 (HIV-1) [41]. A randomized RNA library substituted with the photoreactive chromophore 5-iodouracil was irradiated with monochromatic UV light in the presence of Rev protein. Since modified nucleotides absorb UV light in the 310 nm range, they could be partitioned to be used for the next round of selection. The isolated RNA sequences bound Rev with high affinity [41].

#### 3.2.5. Capillary Electrophoresis-SELEX

Capillary Electrophoresis-SELEX (CE-SELEX) is an alternative SELEX selection procedure developed by Mendonsa and Bowser et al. (2004) [42]. In this method, the randomized nucleic acid library is incubated with the target in free solution. The incubation mixture is injected onto a CE capillary and separated under high voltage, so the nucleic acid–target complex migrates with different mobility than the unbound sequences. The main advantage of CE-SELEX is its high resolving power, because it reduces the number of selection rounds necessary to obtain high-affinity aptamers. Thus, CE-SELEX typically yields high-affinity aptamers after only 2–4 rounds compared to the 6 or 12 rounds required in conventional SELEX [43].

#### 3.2.6. Microfluidic SELEX

Microfluidic SELEX (M-SELEX) was developed by Hybarger G et al. (2006) to synthesize and select a DNA aptamer with high specificity binding to lysozyme [44]. As microfluidic-SELEX can be achieved in a PCR machine, its main advantages are the capability to isolate aptamers with reduced reagent consumption and in fewer rounds of iteration leading to cost reduction in a highly integrated and automated manner [45]. Since its conception, M-SELEX has been improved by combining it with other SELEX technologies such as magnetic bead-based SELEX [46], nanopores [47], capillary electrophoresis [48], and protein microarrays [49].

#### 3.2.7. Magnetic Beads SELEX

Magnetic-bead-based SELEX (MB-SELEX) was developed to select aptamers capable of binding with chloroaromatic compounds, a family of environmental pollutants that were immobilized in magnetic beads [50]. As the target–aptamer complex is separated from the unbound oligonucleotides using magnetic beads, both the elution and precipitation of nucleic acids are not necessary. Magnetic beads with captured templates are added directly to the PCR mixture in amplification rounds, and the aptamers bound to its target are evaluated by fluorescence microscopy [50]. After the original MB-SELEX was developed, several modifications have been made to increase the quantitative capacity of the method. For example, using flow cytometry instead of fluorescence microscopy (FluMag-SELEX) [51].

#### 3.2.8. Toggle-SELEX

Aptamers’ high specificity could be a problem at the moment of preclinical trials. In some cases, oligonucleotides with high affinity and excellent efficiency in vitro fail even before clinical trials because of their lack of compelling efficacy in vivo. The explanation for this is that the SELEX process lacks the whole environmental characteristics present in live organisms [52]. To overcome this, toggle-SELEX was developed. In toggle-SELEX, the aptamers are incubated with one protein for even rounds and another protein or the odd SELEX rounds with the purpose of selecting cross-reactive aptamers capable of binding with both proteins [53].

#### 3.2.9. Cell-SELEX

Proteins are the most common targets in SELEX development; however, purification and maintenance of high purity human proteins in their native conformation is difficult and it is expensive to perform in vitro assays [54]. To solve this problem, Morris, K. N. et al. (1998) developed a method to determine if SELEX could be used with a complex mixture of potential targets in the membranes of human red blood cells [55]. This methodology was called cell-SELEX and allowed the identification of several proteins in its native conformation by using whole cells as the target. Another advantage of cell-SELEX is the identification of target proteins without having prior knowledge of them, making this methodology suitable for the screening of new molecular markers [56]. Cell-SELEX includes positive and negative selection steps (Figure 2). The negative step is needed for removing all the sequences bound to non-desired cells; the positive step allows choosing all the oligonucleotides bound to the target cells [57]. Cell-SELEX has multiple applications such as identification of pathogenic microorganisms [58], biomarker discovery [59], mammalian cells identification [9], cancer diagnosis, and therapy [56,60].

#### 3.2.10. In Vivo SELEX

Although cell-SELEX is a powerful tool used to identify aptamer targets that interact with specific proteins in their native structure, the obtained aptamers are not good enough when used in in vivo models because the aptamer–target binding largely depends on the conformation and, in turn, it is affected by the target’s environment. In 2009, the group of Mi, J. et al. (2009) developed a method to specifically target molecules with the ability to localize hepatic colon cancer metastases in intrahepatic tumor-bearing mice using a large library of nuclease-resistant RNA oligonucleotides [61]. All of the selection process was performed inside the living organism to specifically localize the tumoral cells. In in vivo SELEX, 2′-fluoropyrimidine-modified RNA libraries are injected into the subject and then the aptamers that bind the target tissue are extracted, amplified, and injected into another subject with the same target features [38]. Some examples for in vivo SELEX are aptamers with the ability to localize hepatic colon metastases [61], to penetrate the brain, to act as drug delivery systems for neurological disorders [62], and the capability to be used in antineoplastic therapy against non-small-cell lung cancer [35]. Figure 2 summarizes the process used to obtain aptamers.

## 4. Aptamers in Cancer

Since aptamers can recognize targets and modulate biological activities with higher specificity and affinity than antibodies, it is not surprising that these molecules are considered promising therapeutic tools for the treatment of different types of cancer. Importantly, unlike antibodies, aptamers have low toxicity, are non-immunogenic, and they can easily penetrate the tumor core because of their smaller size [63]. Therefore, a 3-fold increase in the number of articles on aptamers in cancer research has been reported in the last four years (Figure 3A). Between 2012 and 2013 the number of articles using aptamers was less than 100, whereas from 2019 to 2021 it was higher than 300. Notably, more than 22% of the articles reporting the use of aptamers are used in breast cancer, followed by aptamer applications in PCa and lung cancer (Figure 3B and Appendix A).

Numerous works have suggested that aptamers can be used in cancer therapy as inhibitors of growth factors or oncoproteins, as delivery methods for anti-cancer drugs or siRNAs into tumor cells, or even as immune stimulators to fight cancer [64,65,66]; however, most of them are still in a preclinical stage. Noteworthy, the aptamers AS1411 and NOX-A12 are the most advanced aptamer-based therapies for leukemia undergoing clinical research [67]. AS1411 is a guanine-rich 26-base quadruplex DNA aptamer targeting nucleolin (also called C23) [68]. Although nucleolin is found in the cell membrane, it is mainly present in the nucleolus, and high expression of cell surface nucleolin is associated with poor prognosis and higher risk of metastasis [69]. The binding of AS1411 to nucleolin inhibits DNA synthesis that leads to destabilization of BCL2 mRNA and apoptosis [70]. AS1411 is currently undergoing phase II clinical trials for acute myeloid leukemia (AML) and metastatic renal cell carcinoma [54,70,71]. The NOX-A12 aptamer is capable of targeting CXCL12, a chemokine that promotes homing and retention of leukemia cells [71]. Treatment with NOX-A12 sensitizes leukemia cells to conventional therapies and, in combination with bendamustine/rituximab, it improves the therapeutic response in patients with chronic lymphocytic leukemia and multiple myeloma (Table 2) [71].

In addition, the aptamer A30, which binds to the extracellular domain of the human epidermal growth factor receptor-3 (HER3) and does reduce cell proliferation by inhibiting heregulin (HRG) signaling, is under investigation in breast cancer therapy. The combination of A30 aptamer with siRNAs against EEF2, PLK-1, GRK4, and SKIP5, induced specific gene silencing and suppressed cell proliferation. Since the aptamer–siRNA chimera was taken up specifically by HER3-expressing breast cancer cells [72], this aptamer is a promising candidate in breast cancer treatment.

To note, the capacity of aptamers as drug carriers for cancer cells is also a matter of extensive research. Thus, the A10 aptamer that binds to PSMA has been conjugated with doxorubicin to confer both high affinity and specificity against prostate cancer cells. This efficient drug-delivery aptamer significantly inhibited cell proliferation of PSMA-positive cells [73]. Similarly, for AML, aptamer–drug conjugates have been developed. Specifically, aptamer–drug conjugates with methotrexate (Apt-MTX) were able to inhibit AML cell growth, trigger cell apoptosis, and induce cell cycle arrest in the G1 phase in a highly specific manner. Trials with human bone marrow specimens demonstrated that this aptamer–drug conjugate induced selective growth inhibition of primary AML cells without toxicity in normal marrow cells after Apt-MTX exposure. Overall, these findings demonstrate the potential clinical value of Apt-MTX for targeting AML [74]. Another chimeric aptamer siRNA targeting BCL-2 was bound to doxorubicin (siRNA-Dox). siRNA-Dox increased sensitivity of cells to apoptosis and, in turn, decreased cell viability in multi-drug-resistant MCF-7 breast cancer cells [75].

Moreover, aptamers can be used as biosensors for cancer detection. For example, an electrochemical apta-sensor against mucin-1 (MUC1) was recently developed. MUC1 is a surface glycan highly expressed in cancer cells. Voltage changes induced by the chemical reaction between the aptamer conjugated to magnetic beads and gold reduction allow cancer detection by electrochemical analysis [76]. Aptamers can also be used in the field of circulating tumor cells (CTCs). The aptamer BC-15 has been used to specifically identify rare CTCs out of background nucleated cells. This aptamer showed high affinity for nuclei of different human cancer cell lines as well as CTCs isolated from pancreatic cancer patients. The target of the BC-15 aptamer is the heterogeneous nuclear ribonucleoprotein A1 (hnRNP A1). Overexpression of hnRNP A1 has been reported in breast, small cell lung, ovarian, colorectal carcinoma, and pancreatic cancer [77,78,79]. Such results establish a novel way to identify CTCs by using a synthetic aptamer probe [80]. In addition, aptamers can be used in microfluidic systems to capture cells with high affinity. In ovarian cancer, the CX-BG1-10-A aptamer captures CTCs faster than with antibodies in whole blood [81]. Overall, these works show that aptamers are promising candidates to be used in cancer diagnosis and therapy.

## 5. Aptamers in Prostate Cancer (PCa)

Prostate cancer (PCa) is the most frequent cause of cancer-related death in men [82]. The main factors involved in disease etiology are age, lifestyle, and diet. As aging populations worldwide are increasing, the development of new tools to achieve quick and safe diagnosis and treatment in PCa has become highly relevant [83]. The diagnosis and treatment of PCa is a challenge due to the lack of biomarkers and therapeutic targets specific to the disease. For the diagnosis, the PSA-specific prostate antigen is the biomarker used for excellence in the clinic, but it lacks specificity [84,85].

Although most clinicians agree with a PSA threshold of 4.0 ng/mL for men over 50 years old as normal, several factors can produce PSA fluctuations, for example, prostatitis and benign prostatic hyperplasia (BPH) increase PSA levels [86]. Thus, men with PSA levels of 4–10 ng/mL have a 1 in a 4 chance of having PCa, whereas in the cases that PSA is superior to 10, the probability increases to 50% [87,88,89]. Whereas when elevated PSA levels are found, but no symptoms of PCa are present, another PSA test may be recommended to confirm the original finding. If the PSA level is still high, the test must be supplemented with digital rectal exams, imaging tests, and/or prostate biopsy; highlighting the importance of searching for more precise and specific markers in PCa. Hence, other molecules are being studied for the diagnosis of PCa, such as the lncRNA PCA3, a set of kallikreins including klk2 and klk3, and fusion genes such as TMPRSS2-ERG [90,91,92]. Regarding the treatment of PCa, classic chemotherapy consists of blocking androgen receptor activity also called chemical castration. This therapy has good effects at the beginning of the treatment; however, PCa cells become resistant to castration as time progresses [93].

PCa represents a heterogeneous variety of tumors, and numerous studies have reported the use of aptamers in PCa to target these different types of tumors. The yearly average of reported works using aptamers in PCa research has been 28 in the last 10 years (Figure 4A). In these articles, more than 20 proteins have been used as targets in aptamer research in either diagnostic or therapeutic approaches (Figure 4B). In the field of aptamer-based diagnostics, PSA is the most frequently used protein as the target in aptamer technology, whereas PSMA is highly used as the target in therapeutic approaches (Figure 4B).

## 6. Aptamers against Prostate-Specific Membrane Antigen (PSMA)

PSMA is a type II membrane protein expressed in all forms of prostate tissues, including carcinoma [94]. PSMA is coded by *FOLH1*, a gene located at chromosome 11 in a region that is not commonly deleted in prostate cancer. PSMA has peptidase and hydrolase activity and digests dietary folates [95]. As PSMA expression is increased in PCa [96], it has been assayed as a predictor of disease recurrence by using anti-PSMA monoclonal antibodies [97]. In addition, PSMA peptides have been used in PCa treatment for stimulating the immune response by infusing dendritic cells pulsed by these PSMA peptides [98]. Altogether, these works highlight the potential value of PSMA as a target in the development of new approaches to diagnose and treat PCa.

### 6.1. PSMA Aptamers in PCa Diagnosis

Aptamers have been also used as tools for the development of effective diagnostic methods in PCa. For example, an aptamer against PSMA has been used to improve the diagnosis from images by using a nano-crystal semiconductor known as quantum dots (QD’s). Bagalkot et al. developed a targeted QD imaging system with the A10 RNA aptamer capable of differential uptake and imaging of PCa cells expressing PSMA. Some of the advantages of this type of system are wide absorption with narrow photoluminescence, high quantum performance spectra, low photobleaching, and resistance to chemical degradation [99]. Although PSMA is a valuable marker in PCa, two types of cell lines, PSMA (+) and PSMA (−) cells, can be found in the prostate of patients with PCa. Thus, a new approach based on an RNA/peptide dual-aptamer probe was developed by Min et al. (2010) to detect both PSMA (+) and PSMA (−) prostate cancer cells [100]. In this study, two aptamers specific to prostate cancer cells, the A10 RNA aptamer (for the PSMA (+) cell line) and the DUP-1 peptide aptamer (for the PSMA (−) cell line), were conjugated to streptavidin to build the dual-aptamer probe and synchronously detected both prostate cancer cells with a high specificity by electrochemical impedance spectroscopy. Another approach uses lipid nanobubbles functionalized with the A10-3.2 aptamer that, when injected in the abdominal area of mice, the abdominal color Doppler blood flow imaging was significantly improved [101].

In addition, the detection of differentially expressed antigens (biomarkers) has proven to be important for PCa diagnosis and therapy. The main advantage of the use of aptamers over common techniques such as ELISA and tissue staining is that it does not require substantial amounts of starting material. For example, Pai and Ellington adapted the proximity ligation assay (PLA) to cell surface protein targets using modified RNA aptamers detecting and differentiating between cells that distinctively express PSMA tumor antigen [102]. Another application of PSMA aptamers in diagnosis is as biosensors, which are characterized by being ultra-sensitive. Farzin et al. (2017) developed an aptamer-based biosensor (aptasensor) to detect the tumor marker MUC 1 in serum from human samples [103].

### 6.2. PSMA Aptamers in PCa Therapy

The standard care for PCa treatment is prostatectomy or chemical castration to reduce the circulating levels of testosterone and induce the apoptosis of androgen-dependent tumor cells [104]. Although prostate cancer cells are highly sensitive to androgen ablation, prostate cancer tumors contain a population of cells resistant to the treatment, for instance, cells resistant to chemical castration, and prone to both maintenance and progression of the tumor toward metastatic events [105]. Actually, ^177^Lutetium-PSMA-617 has been approved in PCa therapy by the FDA. Although this radioligand therapy has been shown to be safe in patients resistant to chemotherapy, the median progression-free survival is still limited (3.8 months) [106]. Therefore, therapy with aptamers is an advantageous tool since it would allow the design of molecules capable of specifically recognizing various types of tumor populations to achieve an effective pharmacological treatment. Several examples of drug conjugates with aptamers exist in the scientific literature for PCa. Dhar et al. (2011) demonstrated enhanced in vivo pharmacokinetics (PK), tolerability, and efficacy of the cisplatin aptamer (Pt-PLGA-b-PEG-Apt-NP) when compared to cisplatin alone administered in a PSMA-expressing LNCaP subcutaneous xenograft mouse model of PCa [107]. In addition, docetaxel (Dtx), that is the drug of choice in PCa therapeutics, has also been used in aptamer technology to improve its pharmacological properties (administration route, solubility) and decrease its toxicity and side effects. Hence, docetaxel (Dtx)-encapsulated nanoparticles formulated with a biocompatible and biodegradable poly(D, L-lactic-co-glycolic acid)-block-poly (ethylene glycol) (PLGA-b-PEG) copolymer and surface functionalized with the A10 RNA aptamer improved the targeted delivery and uptake of drugs [108]. An additional work by Chen et al. (2016) also showed that the aptamer coupled to nanoparticles and Dtx (Dtx-apt-NPs) improved the antitumor effect in vivo on an LNCaP cell xenograft tumor model and was more effective in inducing LNCaP cell apoptosis or death through G2/M phase cell cycle arrest compared to Dtx-free nanoparticles [109].

Another application of aptamers in PCa has been as vehicles to direct drugs or simply to improve their bioavailability. This approach allows not only a better bioavailability but also a more specific recognition of cancer cells [110]. For example, the use of unimolecular micelles coupled to aptamers as vehicles for transporting doxorubicin to tumor cells of prostate cancer. This type of conjugated molecule induced a high accumulation in the tumor tissue when compared to those without the aptamer in their system [111]. The conjugation of aptamers with liposomes for PCa treatment has been a widely used tool in research. Bandekar et al. (2014) evaluated targeted liposomes loaded with Ac-225 to selectively kill prostate-specific membrane antigen (PSMA)-expressing cells with the aim to assess their potential as targeted antivascular radiotherapy [112].

Moreover, using gold nanoparticles for imaging and therapy of PCa coupled to aptamers has been tested. The latter is based on functionalization of the surface of gold nanoparticles (GNPs) with an RNA aptamer targeting PSMA. The resulting PSMA aptamer-conjugated GNP produced a 4-fold increase in the computed tomography (CT) intensity for targeted LNCaP cells in comparison to non-targeted PC3 cells. Furthermore, the conjugated aptamer was more potent against targeted LNCaP cells than non-targeted PC3 cells when doxorubicin was added to the system [113].

Another interesting application of aptamers is the sensitization of cancer cells to radiotherapy because radio-sensitization can occur by coupling physical agents that allow better absorption of radiation. In this regard, Ni et al. (2011) achieved radio-sensitization of cancer cells with aptamer–shRNA chimeras directed to PSMA for silencing the DNAPK protein [114]. It is well known that approximately 50% of PCa tumors do not express PSMA, some of them because ERG, a common overexpressed transcription factor in PCa, suppresses PSMA expression in tumors containing the TMPRSS2-ERG fusion [90]. To overcome the lack of PSMA, Jing et al. (2016) designed a dual recombinant adenovirus-aptamer system. The viral peptide DUP-1 is capable of recognizing the PSMA-negative cells, while the aptamer A10-3.2 recognizes the PSMA-positive cells. This system decreased the cell growth for both LNCaP (PSMA-positive) and PC3 (PSMA-negative) cells in vitro and in vivo [115].

Although it will be discussed in the aptamer–siRNA chimeras section, it is worth mentioning some examples of these conjugates in PCa therapy. Two anti-PSMA aptamers were designed by Wullner et al., in 2008, with specific cytotoxicity against PCa using siRNA-induced silencing of EEF2, resulting in enhanced cytotoxicity against cancer cells [116]. Dassie et al. (2009) showed that optimized aptamer–siRNA chimeras resulted in regression of PSMA-expressing tumors in athymic mice after systemic administration. This anti-tumor activity was enhanced by increasing the chimera’s half-life using polyethylene glycol [117]. In addition, RNA nanoparticles were constructed by bottom-up self-assembly containing the anti-PSMA aptamer as the targeting ligand and anti-miR17 or anti-miR21 as therapeutic systems. This conjugate was able to strongly bind to the tumors and repressed the tumor growth in mice that received low doses of the conjugate by systemic injection [118]. Finally, an approach that integrates several systems in one is a novel prostate surface membrane antigen (PSMA) aptamer-cationic liposome-double siRNA complex that targets prostate cancer cells to inhibit cell proliferation. The delivery system showed synergism in inhibiting the growth of tumor cells indicating the potential application of the double functional siRNA delivery system for gene therapy in PCa [119].

### 6.3. Chimeras of PSMA

In addition to the traditional use of PCa aptamers in either target recognition or inhibition, aptamers have also been used to mediate targeted delivery of small interfering RNA (siRNAs). This system is called the aptamer–siRNA chimera and allows the delivery of siRNAs in a cell-type-specific manner. McNamara et al. (2006) developed a system based on an aptamer–siRNA chimera whose aptamer portion mediated the binding to PSMA in prostate cancer cells, and the siRNA portion targeted the expression of two survival genes (PLK1 and BCL2) overexpressed in most human tumors [120]. Lupold et al. (2002) characterized two aptamers that bind the extracellular membrane fraction of the PSMA membrane protein [121]. After this, the use of aptamer–siRNA chimeras has been extended to recognize more than one molecular target on PSMA. Mathieu et al. (2017) designed an aptamer–siRNA capable of simultaneously inhibiting EGFR and Survivina [122]. The use of chimeras allows not only the fusion of aptamers with other types of molecules such as siRNAs, liposomes, or viruses but also aptamer–aptamer chimeras are possible. This is the case of the RNA aptamer–aptamer chimera designed for transporting both paramagnetic iron oxide and doxorubicin simultaneously, this chimera being more cytotoxic to the targeted cells [123].

In recent years, the gene editing approaches have gained great importance in the field of cancer research. Besides the use of the well-known CRISPR-Cas system, aptamers both activate and repress gene expression. Li and Li (2017) succeeded in designing a system that fuses a PSMA aptamer with a small activation RNA known as saRNAs [124]. This model promoted the expression of the DPYSL3 protein, decreasing cell migration in cancer cells. Another report shows that pre-treatment of animals bearing PSMA-positive tumors with chemically synthesized and systemically administered aptamer–siRNA chimeras (two days before ionizing radiation therapy) can significantly enhance tumor response to IR [114]. This type of system can be used for different applications as the new multifunctional probe comprising a cell-specific internalization aptamer, fluorescent silver nanoclusters (Ag Ncs), and therapeutic siRNA encompassing one system [124].

Although aptamer–siRNA chimeras have effectiveness in the cell-specific delivery of siRNAs, improvements can be made. One example could contemplate the binding to other peptides that are known to be a fundamental part of the cell or that intervene in some important cell proliferation processes. A biotinylated PSMA-specific aptamer A10 and *SURVIVIN*-siRNA were linked to a Streptavidin-Trans-Activator of Transcription- Double strand RNA binding domain fusion protein (STD protein) to form a therapeutic complex. This complex demonstrated higher efficiency in delivering siRNA into target cells and increasing apoptosis compared to lipofectamine and A10–siRNA chimera [125]. Furthermore, Jiao et al. (2022) developed a ^99m^Tc–Aptamer–siRNA chimera to both diagnose and treat PSMA-positive PCa in vivo [126]. This chimera was composed by the PSMA aptamer A10 to specifically deliver the siRNA against the Mouse double minute 2 homolog (MDM2) in PCa cells. The ^99m^Tc-A10 Aptamer–MDM2 siRNA chimera decreased MDM2 expression in PSMA-positive PCa cell lines. The inclusion of the technetium radionuclide (^99m^Tc) in the chimera allowed a good labeling rate and targeting to the tumor, indicating that the ^99m^Tc-labeled MDM2 siRNA-Apt chimera can be used not only as a nucleic acid treatment drug, but also as an imaging probe [126]. Although further research is necessary, this work demonstrates the potential of aptamers to develop new approaches that integrate diagnosis and treatment of PSMA-positive PCa to provide clinical support to PCa patients.

### 6.4. PSMA Aptamers as Vehicles in PCa

Aptamers can also be used to selectively deliver drugs to PCa cells and enhance their effects at lower concentrations. The use of PSMA aptamers has resulted very efficient for this application because many PCa cells and tissues have high expression of this protein. Dhar et al. (2008) developed cisplatin(IV)-encapsulated nanoparticles with targeting aptamers to effectively deliver cisplatin to PCa cells [127]. These PSMA aptamers showed greater effectiveness (one order of magnitude higher than free cisplatin) of cisplatin on the PCa cell line, LNCaP [127]. This same system has been used to deliver a variety of anticancer drugs such as docetaxel using biocompatible and biodegradable co-polymers such as poly (D, L-lactic-co-glycolic acid)-block-poly (ethylene glycol) (PLGA-b-PEG) functionalized with the A10 aptamer that binds to PSMA [128]. In addition, Singh et al. (2020) encapsulated the A10 RNA aptamer in polysaccharide nanoparticles containing the natural compound thymoquinone (TQ) to inhibit the Hedgehog signaling pathway [129]. The resulting aptamer-based nanoparticles carrying TQ were more effective in both inhibiting the Hh signaling in low drug concentrations and delivering the agent to the PCa cells [129].

PSMA aptamers can further improve their capacity as vehicles when conjugated to liposomal complexes, known as aptosomes. An example for this is an RNA micelle aptamer-conjugated liposome that specifically binds to LNCaP cells expressing PSMA. This aptamosome demonstrated in vivo an anticancer efficacy of the doxorrubicin-encapsulating PSMA-aptamosomes on tumor size regression in LNCaP xenograft mice [110].

The generation of aptamers conjugated to drugs that target PSMA is a growing area of interest in PCa therapy. To mention some seminal research, some involve the co-delivery of shRNAs against Bcl simultaneously with the delivery of doxorubicin. Another example is the encapsulation of cisplatin in positive nanoparticles with PSMA targeting aptamers on the surface of the nanoparticles to specifically deliver cisplatin to PCa cells [127]. The concomitant diagnostic and therapeutic advantages of aptamers have been reflected in the work by Wu et al. (2017), where poly (lactide-co-glycolic acid) nanobubbles (PLGA) modified with the aptamer A-10-3.2 were loaded with paclitaxel, producing tumor regression and diminishing neoplasic characteristics of cells in vitro and in vivo [130]. Table 4 provides an overview of selected PSMA aptamers used for either therapeutic or diagnostic approaches against PCa.

## 7. Other Aptamers in PCa

Although aptamers targeting PSMA are the second most used aptamers in PCa research (Figure 4B), there are others under study and have presented interesting results. Figure 4B and Table 5 summarize reported aptamers in diagnosis and treatment of PCa whose targets are different to PSMA. In the field of aptamers targeting cells, a pluri-targeting DNA aptamer called DML-7 was first designed for recognizing the human PCa cell line DU145. This aptamer internalized into the target cells and exhibited high binding affinity with dissociation constants (Kd) in the nanomolar range. Interestingly, DML-7 bound to DU145 and PC-3 cells but not to LNCaP or 22Rv1 cells [139].

There are many examples of aptamers used as delivery tools. The MUC1 aptamer was explored as a vehicle for delivering doxorubicin to cancer cells. This 86-base DNA aptamer (MA3) bound to the epitope of MUC1 with a Kd= 38.3 nM. The cancer cell lines, A549 (lung) and MCF-7 (breast) express MUC1, the aptamer MA3 preferentially bound to MUC1-positive but not MUC1-negative cells, suggesting that the MUC1 aptamer may have a potential utility as a targeting ligand for selective delivery of cytotoxic agents to MUC1-expressing tumors including prostate [140].

In diagnosis, aptamers have the potential to improve PSA-based tests as they show more sensibility and specificity and are less expensive than ELISA-based tests. The AS1411 aptamer, a 26-base guanine-rich oligonucleotide aptamer, has high affinity to nucleolin on tumor cell surfaces. The AS1411 aptamer labeled with (99m)Tc was stable in normal saline, human serum, and cellular experiments demonstrating specific binding. Since tumors had higher accumulation of radioactivity with this labeled aptamer, it could be a potential tool for use in molecular imaging of PCa [141]. In this line, aptamers have been modified to serve in diagnosis, for example, the mA4 aptamer was modified in the 2′ hydroxyl groups of RNA and poly-T in the 5′ sequence was added to increase its resistance to degradation by nucleases [142]. Other types of aptamers used in the diagnosis of PCa are based on electrochemical detection of PSA. For example, labeled free DNA aptamers coupled to gold particles showed to be useful in PSA detection after electrochemical impedance spectroscopy (EIS) with a range of 1–200 pg mL^−1^. A similar study using a screen-printed carbon electrode (SPCE) showed improved results with a remarkably lower limit of detection of 0.077 pg/mL [143]. Table 5 provides an overview of the main aptamers reported in PCa research with targets different to PSMA.

**Table 5 biomolecules-12-01056-t005:** Aptamers in PCa theranostics.

Target	Application	Main Findings	Type of Aptamer	Sensitivity/Efficacy	Disadvantage/Limitation	References
PCA3	Diagnosis	Aptasensor for the sensitive detection of PCA3	Thioled hairpin DNA aptamer	Linear detection range at 0–150 ng/ml	The establishment of the PCA3 expression depending on the type of prostate cancer cells is needed	[144]
EN2	Diagnosis	Sensitive and specific enzyme-linked oligonucleotide assay (ELONA) for rapid and sensitive detection of EN2 in urine	ssDNA aptamer	EN2-specific (Kd = 8.26 nM) with a limit of detection of 0.34 nM in buffer and 2.69 nM	The capacity for distinguish between their two targets: bladder and prostate cancers	[145]
PSA	Diagnosis	Selective and specific detection of PSA by amperometric electrochemical measurements	A short, single-stranded DNA (ssDNA) pseudoknot forming two stem-loop structural aptamers	Detection range from 10 pg/mL to 500 ng/mL (low detection limit 1.24 pg/mL)	Only control serum samples were used with increasing rPSA concentrations	[146]
Glycosylated PSA and total PSA	Diagnosis	An impedimetric aptamer-based sensor to the dual recognition of PSA	DNA aptamers with binary recognition to the peptide region and the innermost sugar residues	A range between 0.26 and 62.5 ng/mL (PSAG-1)	The work was evaluated using serum samples from men with elevated PSA levels	[147]
Nucleolin	Diagnostic	Highly selective and specific detection of Nucleolin in peripheral blood mononuclear of PCa patients through ELISA assays	DNA aptamer adopting a G-quadruplex structure (AS1411-N5)	High affinity with Kd = 138.1 ± 5.5 nM	Stability of G4 parallel conformation in the presence of other cations	[148]
Neu5Gc	Diagnosis	A sensitive and rapid aptamer-nanoparticle immunochromatographic strip for the visual detection of Neu5Gc was developed	DNA aptamer	The visual limit of detection (LOD) for semi-quantitative detection was 30 ng/mL	Higher LOD than traditional antibody-based ELISA	[149]
PSA	Diagnosis	Detection of label-free, potentiometric detection of PSA with silicon nanowire ion-sensitive field-effect transistor (Si NW-ISFET) arrays	DNA aptamer site-specifically immobilized on Si NW-ISFETs	Concentration-dependent measurements were in a wide range of 1 pg/mL to 1 μg/mL	Does not cover the necessary resolution in the most critical concentration range of ~4 ng/mL	[150]
PSA	Diagnosis	Detection of PSA based on the affinities of the probe aptamer toward Cu-MOG	DNA aptamer-functionalized Cu-MOG	The linear range was from 0.5 to 8 ng/mL, with a detection limit of 0.33 ng/mL	The optimization of several factors as Cu-MOG concentration, time incubation, and Cu-MOG and PA integration is needed	[151]
PSA and VEGF	Diagnosis	Dual biosensor to detect PSA and VEGF released by cancer cells	Thiolated aptamers on gold-covered surface using methylene blue (MB) as redox label	The linear detection ranges (0.08–100 ng/mL for PSA and 0.15 ng–100 ng/mL for VEGF)	Establish the patterns of released proteins by different types of cells to correlate them with cancer aggressiveness	[152]
PSA	Diagnosis	The design of a 2D NS-based PSA aptamer sensor system	DNA aptamer functionalized (MoO3, MoS2, and MoSe2) of two-dimensional nanosheets	The detection limit of PSA was achieved to be 13 pM for MoO3 NSs, whereas the MoS2 and MoSe2 systems exhibited detection limits of 72 and 157 pM	Confocal microscopy assay needed for the in vitro imaging	[153]
ATP, Bcl-2	Treatment	Antiproliferative effect using targeted treatment through antiapoptotic Bcl-2 silencing	Duplex DNA–siRNA chimera	Proliferation and inhibition by inducing apoptosis	Demonstrate the complete lack of toxicity in other normal cells with high production of ATP	[154]
LNCaP cells	Diagnosis and treatment	Targeted drug delivery to treat prostate cancer cells	Doxorubicin loaded DNA aptamer linked myristate chitosan nanogel	Binding affinity above 70%	Demonstrate the effect in tumors with higher diversity of lineage cells	[155]
Vesicle proteins	Profiling study	Improvement in the low abundancy protein analysis in vesicles from plasma and urine samples	SOMAscan	Identification of ~1000 proteins with ~400 proteins present in comparable quantities between plasma and urine vesicles	Standardization of the procedure for obtaining ultrapure vesicles from high proteinous fluids	[156]
PCA3	Diagnosis	Design of novel nucleic acid antibody-like prostate cancer	RNA aptamer	Moderated staining in PCa samples and strong staining in 78% of the cases of BPH	RNA aptamer stability and issues and possible issues in the interaction RNA–RNA	[157]

## 8. Conclusions

The use of aptamers in PCa research currently allows for the improvement of diagnostic tools and antineoplastic drug efficacy. Aptamers also provide a strategy for the recognition of cell markers with high specificity, sensitivity, and efficacy in PCa diagnosis and therapeutics. Therefore, it is not surprising to find a large diversity of aptamers being used in PCa research. In therapeutics, the combination of aptamers with antineoplastic drugs has shown promising results. However, most of the studies are still in a pre-clinical stage so further research in clinical trials with different patient populations is needed to fully demonstrate that aptamers can be successfully used in cancer therapy and diagnosis. In this context, aptamer-based therapies could set a milestone in personalized medicine for PCa treatment given their capacity to recognize cancer cells without affecting non-cancerous ones (Figure 5).

Current PCa markers such as PSA, PCA3, TMPRSS2-ERG, and PSMA are expressed in PCa cells, but some of them are differently expressed depending on the location even in the same tumor from the same patient, complicating the implementation of a general therapy. To overcome this problem, aptamers have been successfully used not only in diagnosis but also in treatment in preclinical studies. Aptamers have been administered as vehicles for drugs to recognize more than one molecular marker in complex systems, making possible the delivery of drugs to different cell types inside the same tumor. These aptamer–drug complexes, often referred to as chimeras, can also include small RNA molecules capable of modulating gene expression. Aptamer–drug chimeras bind to a target on the cell surface, and then they can be internalized and be processed by the silencing machinery to downregulate the expression of specific genes. Overall, aptamers are a promising molecular tool that can be used in the near future for the diagnosis and treatment of several types of cancer, including PCa. The fusion of this type of molecular technology with new generation tools such as sequencing will allow exploring all the advantages of using aptamers for the treatment and diagnosis of PCa.

## Figures and Tables

**Figure 1 biomolecules-12-01056-f001:**
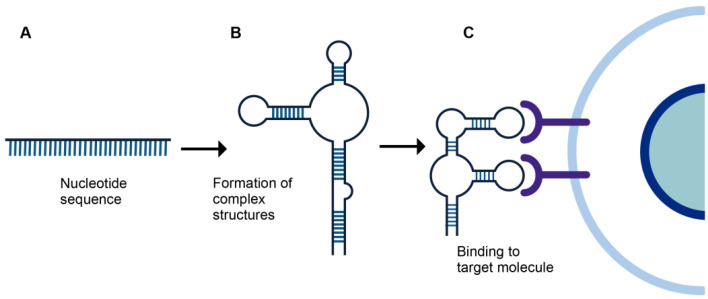
Schematic representation of unfolded, folded, and binding of aptamers to a target molecule. (**A**) Unfolded single strand aptamer. (**B**) Folding of the aptamer by base pairing and tertiary interactions. (**C**) Final structure of the aptamer capable of binding to a specific target molecule.

**Figure 2 biomolecules-12-01056-f002:**
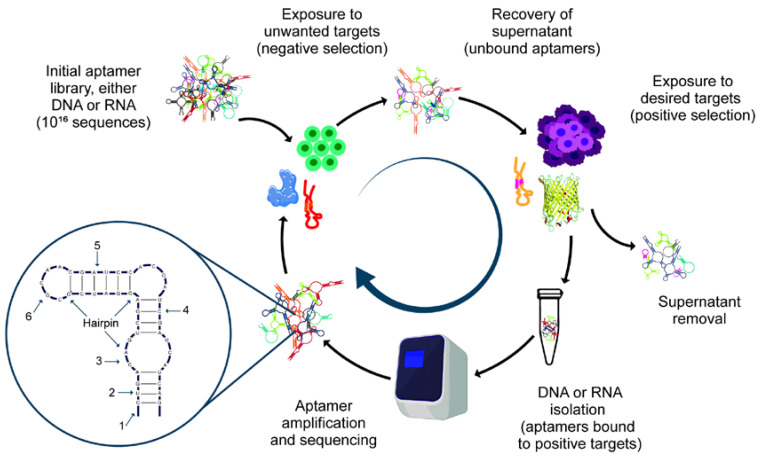
General scheme of the aptamer selection process and possible structural modifications. 1. The 5′ end PEGylation for resisting renal clearance. 2. Nucleobase modification for improving binding affinity and specificity. 3. Phosphodiester backbone modifications for resisting nuclease degradation. 4. Modifications on the sugar ring for resisting nuclease degradation. 5. The 3′ end-capping strategy resisting nuclease degradation. 6. The mirror image L-deoxyoligonucleotide resisting nuclease degradation. In the cell-SELEX process, green cells represent the non-desired cells used for the negative selection step and magenta cells represent the target cells required for the positive selection.

**Figure 3 biomolecules-12-01056-f003:**
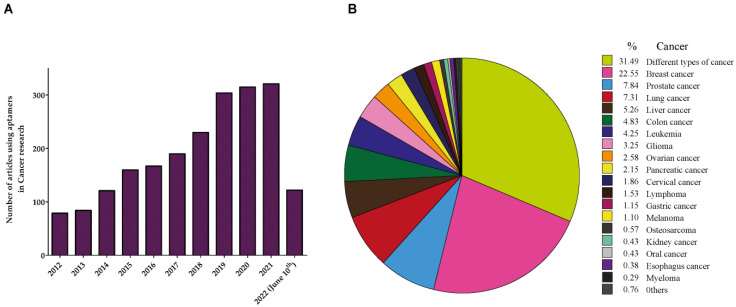
Aptamers in cancer research. (**A**) Yearly trend of published articles reporting the application of aptamers in cancer research. (**B**) Proportion of types of cancer in which aptamers are most frequently used for therapeutic or diagnostic purposes.

**Figure 4 biomolecules-12-01056-f004:**
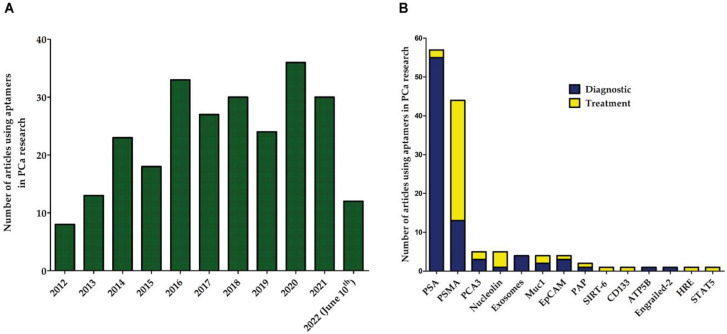
Aptamer publications in PCa research. (**A**) Yearly trend of published articles reporting the use of aptamers in PCa. (**B**) Proteins used as targets of the aptamers in PCa research. Blue bars represent the number of articles using aptamers directed against each protein in diagnosis, whereas yellow bars refer to articles using aptamers in PCa therapy. PCA3 = prostate-cancer-associated 3; Muc1 = mucin 1; EpCAM = epithelial cellular adhesion molecule; PAP = prostatic acid phosphatase; SIRT-6 = sirtuin 6; CD133 = prominin-1; ATP5B = ATP synthase F1 subunit beta; HRE = hormone response element; STAT5 = signal transducer and activator of transcription 5A.

**Figure 5 biomolecules-12-01056-f005:**
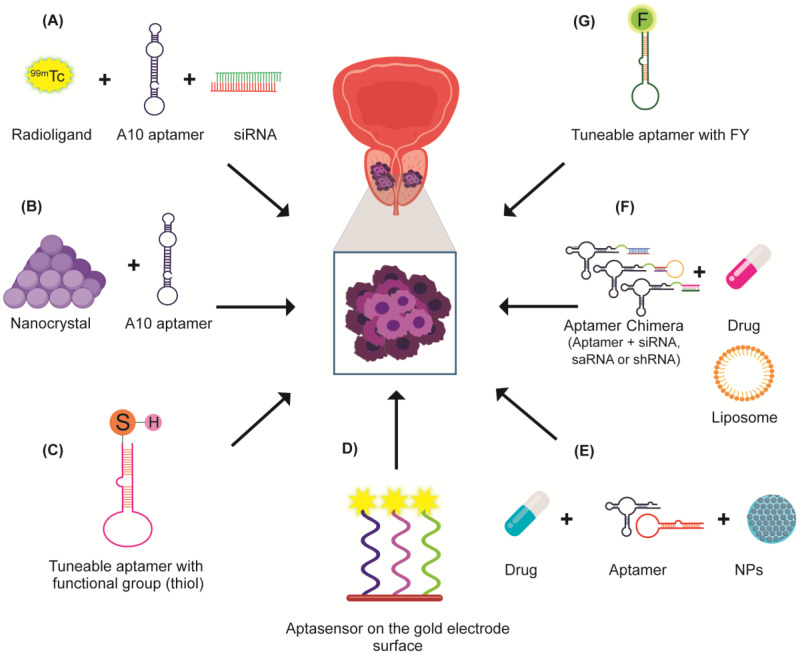
Scheme depicting some applications of aptamers in the diagnosis and treatment of PCa. (**A**) Aptamers can be coupled to radioligands and siRNA to specifically deliver the siRNA in PSMA^+^ cells. The inclusion of the technetium radionuclide (^99m^Tc) allows the use of this aptamer in either therapeutic or diagnostic purposes. (**B**) Aptamers may improve the diagnosis imaging by implementing the semiconductor and nanocrystals technology known as quantum dots. (**C**) Aptamers can be modified with thiol groups to give them greater stability. (**D**) Aptamers can be used as aptasensors on gold electrode surfaces to recognize molecular targets. (**E**) Conjugates with aptamers are anchored to nanoparticles for systemic delivery of drugs. (**F**) Aptamers are used in therapy as chimeras apt-siRNA, apt-saRNA, apt-shRNA, apt-drugs, or apt-liposome by directly targeting the tumor. (**G**) Aptamers can be modified with 2′ fluoropyrimidine to increase stability.

**Table 2 biomolecules-12-01056-t002:** Ongoing clinical trials using aptamers in either diagnosis or treatment of several diseases.

Aptamer	Target	Phase	Patient Number	Goal	Clinical Trial Identifier
N/A	COVID-19	Recruiting	200	COVID-19 test	NCT04974203
ARC1905	C5	I	47	Macular degeneration	NCT00950638
E10030	PDGF	II	449	Neovascular age-related macular degeneration	NCT01089517
APT-POCT-01	Antiretrovirals	Completed	30	Test the adherence to antiretrovirals	NCT04302896
Sgc8	PTK7	Unknown	70	Treatment of colorectal cancer	NCT03385148
REG1	IXa	I	107	Anticoagulation system	NCT00113997
EYE001	VEGF	I	5	Reduce retinal thickening and improve vision in patients with Von Hippel–Lindau syndrome (VHL)	NCT00056199
Oxytocin aptamer	Oxytocin	Unknown	28	Test novel aptamer-based electrochemical assay for the detection and quantification of salivary oxytocin	NCT03140709
ARC1779	Not specified	II	28	Treatment of Von Willebrand factor-related platelet disorders	NCT00632242
ApToll	TLR4	I	46	Stroke	NCT04742062
NOX-H94	Hepcidin	II	33	Treatment of anemia of chronic disease	NCT02079896
NOX-E36	CCL2	I	72	Chronic inflammatory diseases	NCT00976729
NOX-A12	CXCL2	I	48	To fight solid tumors by modulating the tumor microenvironment	NCT00976378
ApToll	COVID-19	Recruiting	30	Block the progression of patients to cytokine storm syndrome (CSS)	NCT05293236
AS1411	Nucleolin	II	90	Treatment of patients with primary refractory or relapsed acute myeloid leukemia	NCT01034410

**Table 3 biomolecules-12-01056-t003:** The main tools used for the design of RNA aptamers in silico.

Tool	Type	Description
MPBind	Ranking	Ranks aptamers according to a statistical score. It is based on four types of Z score for each sequence motif (k-mer).
FASTAptamer	Clustering	Clusters aptamers based on sequence analysis.
MEMERIS	Motif	Finds motifs for its secondary structure and predicts the sequence motif in the loop structure.
AptaMut	Optimization	Determines whether mutations contribute to increase the affinity of the aptamers relative to the parent sequence.
Rtools	Others	Analyzes secondary RNA structures from a simple sequence of nucleotides.
COMPAS	All	Performs quality control, aptamer identification, ranking, clustering, and optimization.

**Table 4 biomolecules-12-01056-t004:** Overview of PSMA aptamers and applications.

Application	Designing Method	PSMA-Aptamer Sequence and Modifications	Sensitivity or Results	Disadvantage/Limitation	Biological Target	References
Diagnosis	Chemical synthesis	NH2-GAATTCGCGTTTTCGCTTTTGCGTTTTGGGTCATCTGCTTACGATAGCAATGCT	~100 particles/μL in urine	Improvement in the number of particles per microliter to overcome current analysis	Urine	[131]
Diagnosis	Chemical synthesis	NH2-GGGAGGACGAUGCGGAUCAGCCAUGUUUACGUCACUCCU	Detection concentration in vitro is 10 μg/mL	Needs to demonstrate the distribution in other tissues apart of the pulmonary system	Cell lines and mice xenografts	[132]
Diagnosis	Chemical synthesis	Cy5.5-GGGAGGACGAUGCGGAUCAGCCAUGUUUACGUCACUCCU-spacer-NH2-3′ with 2′-fluoro pyrimidines	AUC of ½ peak intensity (dB·s) is 1507.60 ± 269.33	Characterization of the nanobubbles’ distribution and their elimination is needed	Cell lines and mice xenografts	[101]
Treatment	Cell-SELEX	PSMA aptamer-survivin antisense siRNA: GGGAGGACGAUGCGGAUCAGCCAUGUUUACGU CACUCCUAAAAUGUAGAGAUGCGGUGGUCCUU	Inhibition of tumoral growth in mice	RNA stability and siRNA dosage for keeping therapeutic effect without toxicity in other tissues	Cell lines and mice xenografts	[133]
Diagnosis and prognosis	Chemical synthesis	Acid aptamer A10-3.2 (no specified sequence or modifications)	Discrimination of prostate cancer cells that express PSMA	Adjustment in parameters as the intensity of light penetration to deep tissue, and TMIA-chromophore abundance due to target density from small tumors	Cell lines	[134]
Treatment	SELEX	A10-3.2-saV2-9: TAA TAC GAC TCA CTA TAG GGA GGA CGA TGC GGA TCA GCC ATG TTT ACG TCA CTC CTA gaa aga aca tga atg ctg c ATGAAGCTTG g cag cat tca tgt tct ttc dTdT	Adjunctive therapy to suppress prostate cancer metastasis	Needs to demonstrate the specific gene activation by saRNA	Cell lines and mice xenografts	[135]
Treatment	Chemical synthesis	GGGAGGAAUAGCUGACGGGAGGACGAUGCGGAUCAGCCAUGUUUACGUCACUCCUUGUCAAUAAUAAGGGGC	Cytotoxicity in prostate cancer cells	Needs characterization of the in vivo stability for the biotin–DNA linker	In silico modeling and cell lines	[123]
Treatment	Chemical synthesis	GGGAGGACGAUGCGGAUCAGCCAUGUUUACGUCACUCCUUGUCAAUCCUCAUCGGC	Enhancement in the potency of external beam radiation therapy for established PSMA-positive tumors	Possibility of aptamer–siRNA chimera-mediated inflammatory reactions in humans	Cell lines, mice xenografts, and tissue sections	[114]
Treatment	Chemical synthesis	TGX221 with PSMA aptamer conjugation (no specified sequence or modifications)	Effective anti-cancer agent for prostate cancer	Possible accumulation of the nanomicellar compounds in tissues	Cell lines, mice xenografts and tissue sections	[136]
Treatment	Chemical synthesis	GGGAGGACGAUGCGGAUCAGCCAUGUUUACGUCACUCCU-(CH2)6-S-S-(CH2)6-OH-3′ with 2′-fluoro pyrimidines	Efficient delivery of miRNA expression vectors to prostate cancer cells	Needs the determination of toxicity in other tissues in vivo	Cell lines and mice xenografts	[137]
Diagnosis and treatment	Chemical synthesis	GGGAGGACGAUGCGGAUCAGCCAUGUUUACGUCACUCCUAA	To diagnose and treat PSMA-positive PCa in vivo	Determination of the possible undesired effects of 99m Tc radiolabeled aptamer	Cell lines and mice xenografts	[126]
Diagnosis	Chemical synthesis	DBCO-5′-GAA TTC GCG TTT TCG CTT TTG CGT TTT GGG TCA TCT GCT TAC GAT AGC AAT GCT-3′	Diagnostic potency (AUC: miR-145, 0.76; miR-221, 0.7; miR-451a, 0.65; and miR-141, 0.64)	Issues with the high relative amount of PSMA(+)SEVs observed in plasma	Blood samples	[138]

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
