# Peer review of "Aptamers as Theragnostic Tools in Prostate Cancer"

_biomolecules, 2022, doi:10.3390/biom12081056_

Round 1

Reviewer 1 Report

Manuscript ID: biomolecules-1793035

General comments:

The present review entitled “Aptamers as theragnostic tools in Prostate Cancer” aimed to provide an overview aim of this review was to compile the existent scientific literature regarding the use of aptamers in prostate cancer in diagnosis and treatment. 

However, the authors should consider recent puplications, with higher scientific evidence, namely meta-analyses. 

The manuscript should be revised in order to the main content. Some topics should be revised taking into consideration the provided information, namely the excess of information, already provided by other authors in this field. 

Author Response

Dear Reviewer,

We really appreciate the comments made to the manuscript. We accepted your suggestions nd a point-by-point report is provided.  We strongly believe that these modifications have improved the manuscript.

General comments:

The present review entitled “Aptamers as theragnostic tools in Prostate Cancer” aimed to provide an overview aim of this review was to compile the existent scientific literature regarding the use of aptamers in prostate cancer in diagnosis and treatment. 

However, the authors should consider recent puplications, with higher scientific evidence, namely meta-analyses. 

Response: This suggestion is very appropriate, therefore 3 tables (Table 4 table 5 and supplementary Table 1, pages 20 and 22, respectively) were added to the manuscript that include recent publications (from 2018 to present date) of aptamers in Prostate Cancer.

The manuscript should be revised in order to the main content. Some topics should be revised taking into consideration the provided information, namely the excess of information, already provided by other authors in this field. 

Response: Following reviewer´s suggestion, the manuscript was thoroughly revised to remove unnecessary or repetitive information. In addition, since a new table dealing with the aptamers used in Prostate Cancer was included, only the most relevant works remained in the text.

Reviewer 2 Report

The Review by Cruz-Hernandez is covering a large scope concerning the use of aptamers as a theranostic tool in prostate cancer.

As a general comment, the Review is interesting for the field and fits the scope of the journal. Though aptamers have been described quite a while ago, they could indeed represent interesting alternatives for both diagnosis or/and treatment of PCa.

I think however that both in its present form, quite a number of issues that are detailed below need to be addressed prior to acceptance.

- First and most predominantly, the way the manuscript is written gives the impression that aptamers are very efficient both in terms of therapeutic potential and/or delivery of drugs to PCa tumor cells both in vitro and in vivo and that this strategy if quite efficient as compared to other drugs that are routinely used in the clinic. However, this is definitely far from the truth as the authors point out in the conclusion. I think that this enthusiasm for aptamers should be tempered down throughout all the manuscript and that results that have been obtained should be restrained to the models that were used for it. Indeed, there is no aptamer FDA approved despite the long-lasting research that was performed with aptamers. This is in deep contradiction with immune checkpoint inhibitors or targeted therapies (to take these two examples).

- Introduction, line 41 =, please explicit what is directed therapeutic agents.

- line 44, the sentence is too long. Moreover, failure to develop these agents is not only due to toxic effects but also to the absence of effect in vivo….

- line 53 , please define PCa and PSMA for the first time and explain the role of the protein;

- Line 74, while I understand the optimism of the authors for aptamers, they only describe advantages as compared to antibodies …..To play devil’s advocate, I could say that many antibodies have been approved and are therapeutically active while no aptamer has gone through phase III yet. The authors should therefore mention disadvantages of aptamers to be fare…

- Line 88, please explain in more details what Pegaptanib is ?

- Line 174, please rephrase the first sentence as it is unreadable.

- Line 195, please explain what a anti-lzozyme aptamer is for non-experts and how it works.

- Line 222: This section is rather clear as it is explained but maybe a scheme would be helpful to better clarify what is Cell-SELEX

- Line 277, the authors mention that” aptamers are used in cancer therapy”. This is untrue as none of them are approved (at least to my knowledge). Actually authors further mention few aptamers in clinical trials in the same paragraph.

- Line 293, please explicit HRG

- Line 317 and further, the authors mention the use of aptamers for the detection of CTCs to “target” them…I am actually confused by this term…Is the BC-15 used for killing CTCs and in that case how this is occurring? (what is BC-15 recognizing?) or is it used to detect CTCs and can give access to a count of these tumor cells? Please clarify.

- Line 332, concerning PSA levels and PCa, the authors should be more precise regarding the actual value of PSA marker. Indeed, the level per se cannot be indicative of a cancer, but high PSA levels may not either as it could be due to inflammation. Actually elevated PSA or increasing values of PSA when it is measured twice with 3 weeks interval is more indicative of a potential tumor growth…Please clarify.

- Line 335-336, the TMPRSS2-ERG (or ETS in general) fusion genes represent only a subset of PCa.

- Line 346: What “whereas PSMA is highly used as target in approaches” means?? Please clarify

- Line 352, Should read PCA3 not PCAC3

- Lines 370: the paragraph should deal with PSMA but presents PSA concentration measurments using aptamers? The authors should modify the title in accordance. In this line, why is it so important to be so sensitive for the PSA detection? Is there any clinical rationale to reach a detection limit of 0.032 ng/ml? Please explain?

- Line 390, the authors statet that to overcome the limitation that PSMA aptamer do not detect PSMA negative cells, dual RNA/peptide aptamers probes have been developed. However it is really difficult to understand how this works for non-experts as it is not explained or very confusing.

- Line 403, what is the difference between aptamer-sensor and aptamer ? please give a better description of the difference.

- Page 13, a part of the Ms is dedicated to aptamers used to deliver chemos in PCa cells. While the studies that are mentioned are indeed interesting as they show a better effect of the aptamer strategy, they used cisplatin or doxorubicin that are not really the drug of choice routinely used in the clinic. It would be interesting to discuss these choices and implement with other studies using docetaxel or cabazitaxel if they exist.

- Line 498: Please explicit what is a STD protein

- Page 15, paragraph 6.4 is partly duplicating what has been review earlier. Please reconcile.

- Lines 536-539, should it be aptosome or aptamosome?

- Page 16, section 7, addresses aptamers targeting cells which is indeed interesting. However, I think that this paragraphs lacks molecular explanations on how these aptamers are targeting cells? What is (are) the target(s)? This section needs more explanations, especially for non-experts.

- The conclusion is fine as it is (for the first time) acknowledging the fact that, despite all these studies, aptamers are still being tested in clinical trials.

Author Response

Dear Reviewer,

We really appreciate the time spent for reviewing our work as well as all the suggestions made to the manuscript. We accepted most of your comments and a point-by-point report is provided below. We strongly believe that these modifications have improved the manuscript.

PLEASE SEE THE ATTACHMENT.

Reviewer 3 Report

Cruz-Hernandez and colleagues wrote an excellent review about Aptamers in prostate cancer. Therefore, I recommend acceptance after minor revision. However, the following issues must be improved, and additional information needs to be added:

• 177Lutetium-PSMA-617 has been approved as therapy by the FDA. Moreover, Prostata PET-CT with radioactive labelled PSMA is also performed regularly. Please add a paragraph with the advantages/disadvantages of aptamers compared to the already established theranostics.

• Please add an overview table about the ongoing clinical trials to the manuscript similar to Table 2 of Ebersbach et al. (https://doi.org/10.3390/cancers13194854)

• In 2018 a review about PSMA aptamers was published by Lupold et al. (PMCID: PMC5902725), introducing several PSAM aptamers (Table 1). In line with this review, please add an overview table with sequences and characteristics of available PSMA aptamers, especially those developed since 2018.

Author Response

Dear Reviewer,

We appreciate the time spent for reviewing our work as well as all the suggestions made to the manuscript. We accepted most of your comments and a point-by-point report is provided below. We strongly believe that these modifications have improved the manuscript.

Cruz-Hernandez and colleagues wrote an excellent review about Aptamers in prostate cancer. Therefore, I recommend acceptance after minor revision. However, the following issues must be improved, and additional information needs to be added:

  • 177Lutetium-PSMA-617 has been approved as therapy by the FDA. Moreover, Prostata PET-CT with radioactive labelled PSMA is also performed regularly. Please add a paragraph with the advantages/disadvantages of aptamers compared to the already established theranostics.

Response: Following reviewer´s comment, a paragraph with the advantages of aptamers compared to therapy with 177Lutetium-PSMA-617 was included in lines lines 488-489.

  • Please add an overview table about the ongoing clinical trials to the manuscript similar to Table 2 of Ebersbach et al. (https://doi.org/10.3390/cancers13194854)

Response: Table 2 (Page 4-5) was added to the manuscript describing the ongoing clinical trials dealing with the use of Aptamers.

  • In 2018 a review about PSMA aptamers was published by Lupold et al. (PMCID: PMC5902725), introducing several PSMA aptamers (Table 1). In line with this review, please add an overview table with sequences and characteristics of available PSMA aptamers, especially those developed since 2018.

Response: This suggestion is very appropriate; therefore Table 4 was included to provide more information of the recent literature using PSMA aptamers in therapeutics and diagnostics of Prostate Cancer. This table provides information about the use of the aptamer, its sensitivity, sequence, limitation and the biological target used in each work.

Reviewer 4 Report

The manuscript by Cruz-Hernàdes and co-authors focuses on the clinical application of aptamers.

The review is well written and organized.  It is a landscape of the biology of aptamers, the selection procedures, and the clinical applications in cancers,  with a particular focus on prostate cancers.

I have a few suggestions.

The authors have to include a table summarising the overall aptamers used in cancers.

The authors must include a table summarising the aptamers used in prostate cancer therapy.

Author Response

Dear Reviewer,

We appreciate all the comments made to the manuscript. We accepted most of your comments and a point-by-point report is provided below. We strongly believe that these modifications have improved the manuscript.

The manuscript by Cruz-Hernandez and co-authors focuses on the clinical application of aptamers.

The review is well written and organized.  It is a landscape of the biology of aptamers, the selection procedures, and the clinical applications in cancers, with a particular focus on prostate cancers.

I have a few suggestions.

The authors have to include a table summarizing the overall aptamers used in cancers.

Response: Thank you for the suggestion. Although Figure 3 (Page 11) gives an overview of the overall aptamers used in cancers, following your suggestion, supplementary Table 1 was added to the manuscript (Page 10) to provide a detailed description of the aptamers used in cancer in 2022 (name of the work, purpose (diagnostic or therapeutic), cancer type). Besides, as the number of published articles from 2018 to 2022 is higher than 1200, providing such table in the manuscript was out of the scope of the present work. A pdf file that provides both Supplemtary Table 1 (Pages 1-3) and the information of such works (Pages 4 - 9) is provided as an attached file and it is to your consideration whether it would be included in the manuscript as supplementary material.

The authors must include a table summarizing the aptamers used in prostate cancer therapy.

 Response: We agree with your suggestion, and Table 5 was incorporated to the manuscript describing the aptamers used in Prostate Cancer. This new table provides information about application, use, sensitivity/specificity, disadvantages/limitations of each aptamer described in this table.

Round 2

Reviewer 2 Report

Drastic changes have been made in accordance to the Reviewer's remarks.  Additional helpful informations have been added to the review and are really strengthening the manuscript. I would recommend acceptance in its present form.